# Impact of Previous Occupational Exposure on Outcomes of Chronic Obstructive Pulmonary Disease

**DOI:** 10.3390/jpm12101592

**Published:** 2022-09-27

**Authors:** Youlim Kim, Tai Sun Park, Tae-Hyung Kim, Chin Kook Rhee, Changhwan Kim, Jae Seung Lee, Woo Jin Kim, Seong Yong Lim, Yong Bum Park, Kwang Ha Yoo, Sang-Do Lee, Yeon-Mok Oh, Ji-Yong Moon

**Affiliations:** 1Division of Pulmonary and Allergy, Department of Internal Medicine, Konkuk University Medical Center, Konkuk University School of Medicine, Seoul 05030, Korea; 2Division of Pulmonology, Department of Internal Medicine, Hanyang University Guri Hospital, Hanyang University College of Medicine, Guri 11923, Korea; 3Department of Internal Medicine, Seoul St. Mary’s Hospital, Catholic University of Korea, Seoul 06591, Korea; 4Department of Internal Medicine, Jeju National University Hospital, Jeju National University School of Medicine, Jeju 63241, Korea; 5Department of Pulmonary and Critical Care Medicine, Asan Medical Center, University of Ulsan College of Medicine, Seoul 05505, Korea; 6Department of Internal Medicine, College of Medicine, Kangwon National University, Chuncheon 24341, Korea; 7Division of Pulmonary and Critical Care Medicine, Department of Medicine, Kangbuk Samsung Hospital, Sungkyunkwan University School of Medicine, Seoul 03181, Korea; 8Department of Pulmonary and Critical Care Medicine, Hallym University Kangdong Sacred Heart Hospital, Seoul 05355, Korea

**Keywords:** occupational exposure, chronic obstructive pulmonary disease, acute exacerbation, lung function

## Abstract

Occupational exposures have been regarded as a risk factor for the development of chronic obstructive pulmonary disease (COPD). However, there is little knowledge regarding the effect of occupational exposure on the treatment outcomes of COPD. Therefore, the aim of this study was to evaluate the question of whether occupational exposure can have a potential impact on COPD outcomes. **Methods:** Information regarding self-reported occupational exposure for 312 patients with COPD from the Korean Obstructive Lung Disease (KOLD) Cohort were included. A comparison of the rate of acute exacerbation, annual lung function change, and quality of life according to the presence or absence of occupational exposure was performed. **Results:** Seventy-six patients (24.4%) had experienced occupational exposure; chemical materials were most common. At enrollment, a higher COPD-specific version of the St. George Respiratory Questionnaire total score (39.7 ± 18.8 vs. 33.1 ± 17.6, *p* = 0.005) and a higher exacerbation history in the past year (30.3% vs. 17.5%, *p* = 0.017) were observed for patients with occupational exposure compared to those without occupational exposure. Furthermore, in the follow-up period, after adjusting for potential confounders, a higher frequency of acute exacerbation (odd ratio, 1.418; 95% confidence interval, 1.027–1.956; *p* = 0.033) and a more rapid decline in forced expiratory volume in 1 s (*p* = 0.009) was observed for COPD patients with occupational exposure compared to those without occupational exposure. **Conclusions:** In the KOLD cohort, worse outcomes in terms of exacerbation rate and change in lung function were observed for COPD patients with occupational exposure compared to those without occupational exposure. These findings suggest that occupational exposure not only is a risk factor for COPD but also might have a prognostic impact on COPD.

## 1. Introduction

Chronic obstructive pulmonary disease (COPD) is characterized by persistent limitation of airflow associated with a chronic inflammatory response in the airways to noxious particles or gases [1]. Cigarette smoking, which induces chronic inflammation of the airways and lungs, is a well-known risk factor [2,3]. However, exposure to smoking would obscure the possible role of environmental factors in chronic airway inflammation or the development of COPD [4]. In particular, exposure to smoking does not fully explain COPD in non-smokers. According to a recently published paper, occupational and environmental exposures are regarded as an additional risk factors for the development of COPD [4]; in particular, it is estimated that 19.2% of COPD in smokers and as much as 31.1% in never smokers can be attributed to occupational exposure to dust, chemical agents, gases, and fumes [5].

Previously reported studies mainly focused on the association between cigarette smoking and the disease course of COPD; thus, there is little knowledge regarding the impact of occupational exposure on the prognosis of COPD [6]. Although a few studies focusing on a causal relationship between occupational exposure and the development of COPD have been reported [5,7,8,9,10,11,12,13,14], it is uncertain whether occupational exposure to noxious particles can affect the clinical features and prognosis of COPD. Thus, studies examining the effect of occupational exposure on COPD-related outcomes, i.e., acute exacerbations, lung function change, and quality of life of COPD patients, are needed.

Therefore, using the longitudinal observational cohort study, the aim of this study was to estimate the relationship between occupational exposure and clinical outcomes of COPD, including acute exacerbations, lung function change, and quality of life.

## 2. Methods and Materials

### 2.1. Study Design and Participants

The Korean Obstructive Lung Disease (KOLD) cohort is a multicenter, prospective longitudinal cohort designed for the development of new classification models and biomarkers for use in the prediction of clinically relevant outcomes for patients with obstructive lung disease; details of the KOLD study were previously published [15]. Subjects were recruited from pulmonary clinics in 16 hospitals in the Republic of Korea from June 2005 to April 2012. The inclusion criteria for this study were as follows: (1) subjects with COPD, as defined by a post-bronchodilator (postBD) forced expiratory volume in one second (FEV_1_)/forced vital capacity (FVC) less than 0.7 and more than ten pack-years of smoking history; (2) those without history or radiographic evidence of tuberculosis, bronchiectasis, or other pulmonary disorder; and (3) those with answers to the question regarding absence or presence of occupational exposure. In addition, subjects who had experienced COPD exacerbations within two months were excluded.

Demographic and clinical data including age, sex, body mass index, smoking status, and exacerbation in the previous year were collected at the time of enrollment in the study. Computed tomography (CT) was performed, and the calculation of the emphysema index was performed using the CT image. Measurement of patient-reported outcomes was also performed. Evaluation of the degree of dyspnea was performed according to the modified Medical Research Council (mMRC) dyspnea scale [16]. Assessment of health-related quality of life was performed using the COPD-specific version of the St. George Respiratory Questionnaire (SGRQ-c) [17] and COPD Assessment Test (CAT) [18]. Assessment of comorbidities was also performed using the modified Charlson comorbidity index (mCCI) [19] and the BODE index [20].

The study protocol was approved by the institutional review boards of Asan Medical Center (Approval No. 2005–0010) and 16 other hospitals. Informed written consent was obtained from all patients.

### 2.2. Occupational Exposure

History of occupational exposure was assessed by questionnaire as follows: “Have you ever been exposed to the following substances in your work environment?: Isocyanate, toluene di-isocyanate, welding solvent, wheat flour powder, grain dust, hair permanence drug, and other chemical substances.” Subjects were categorized according to exposure history: group with occupational exposure vs. group without occupational exposure.

### 2.3. Spirometry and Six-Minute Walk Test

Spirometry was performed according to the recommendations of the American Thoracic Society (ATS)/European Respiratory Society (ERS), using a Vmax 22 (Sensor-Medics, Yorba Linda, CA, USA) or a PFDX (MedGraphics, St. Paul, MN, USA) [21]. The percentage predicted values (% predicted) for FEV_1_ and FVC were calculated from equations obtained in a representative Korean sample [22]. Diffusing capacity for carbon monoxide (DLco) was measured by the single-breath method using a Vmax229D (Sensor-Medics) or a Masterlab Body (Jaeger AB, Würtsburg, Germany), following recommendations of the ATS/ERS protocol [23]. The predicted values of DLco were calculated from Park’s equation formulated using data from a healthy Korean population [24]. The six-minute walking distance test (6MWD) was performed according to the ATS guideline [25].

### 2.4. Outcomes

According to the guideline of the Global Initiative for Chronic Obstructive Lung Disease (GOLD), an acute exacerbation was defined as worsening respiratory symptoms, requiring systemic steroids and antibiotics, a visit to the emergency room, or admission to a hospital [26]. Interviews with patients were conducted every three months during the follow-up period using the pre-structured interview sheet, which includes the question of whether they had visited the outpatient clinics or emergency rooms due to an increased amount of sputum, purulent sputum, or deterioration of dyspnea within three months. Exacerbation was documented at each hospital and traced using medical records. If the patients visited other clinics or emergency rooms, the name of clinics or emergency rooms, the reason for the visit, date of visit, and frequency were collected through the pre-structured interview sheet.

### 2.5. Statistical Analyses

Statistical analysis was performed using SPSS version 20.0 (SPSS Inc, Chicago, IL, USA). All values were expressed as mean ± standard deviation or *n* (%). The χ^2^ test and Fisher’s exact test were used for categorical variables, and the student’s *t*-test or Mann–Whitney *u*-test was used for continuous variables. Logistic regression analysis for repeated measures using generalized estimating equations (GEE) was performed for construction of models for risk factors of acute exacerbation of COPD. An assessment of factors that impact the annual change in lung function (postBD FEV_1_) and quality of life (SGRQ-c) was performed using a linear mixed model (LMM). *p*-values <0.05 were considered to indicate statistical significance.

## 3. Results

### 3.1. Baseline Characteristics of the Study Population

Of the participants in the KOLD cohort, 312 subjects with COPD were identified and 76 patients (24.4%) were classified as having had occupational exposure (Figure 1a). A list of specific occupational exposures of 76 patients is shown in Appendix A Table A1; the agents with the most common exposure were chemical materials, including isocyanate, toluene di-isocyanate, and welding solvent. The baseline characteristics of enrolled patients according to occupational history are shown in Table 1. Patients who had experienced occupational exposure had significantly lower levels of education (*p* = 0.012) and quality of life (higher score of SGRQ-c, 39.7 ± 18.8 vs. 33.1 ± 17.6; *p* = 0.005) compared to those without occupational exposure. However, there was no difference in health-related outcomes according to CAT score and degree of dyspnea (*p* = 0.316 and 0.105, respectively). In addition, no specific findings for other variables, including age, sex, percentage of current smokers, body mass index (BMI), exercise capacity (6MWD), BODE index, mCCI, spirometric values, and emphysema index were observed between the two groups (Table 1).

At enrollment, patients with occupational history had experienced more exacerbations in the previous year (30.3% vs. 17.5%, *p* = 0.017). While both groups were similar in terms of follow-up periods (54.2 ± 22.5 vs. 56.5 ± 24.5, *p* = 0.482) and the presence of exacerbation during the follow-up period (69.6% vs. 67.6%, *p* = 0.758) (Table 1), a higher annual exacerbation rate after enrollment was observed for COPD patients who had experienced occupational exposure (0.65 ± 0.66 vs. 0.42 ± 0.49, *p* = 0.038) (Figure 1b).

### 3.2. Occupational Exposure as a Risk Factor for Acute Exacerbation of COPD

The logistic regression analysis using GEE for risk factors for acute exacerbation of COPD is shown in Table 2. In model 1, history of occupational exposure, sex, age, BODE index, smoking status, and history of exacerbation during the previous year were considered risk factors, and BODE index, history of previous exacerbation, and occupational exposure were significant for future exacerbation (BODE index, odds ratio (OR) = 1.147, 95% confidence interval (95% CI) = 1.065–1.235, *p* < 0.001; history of previous exacerbation, OR = 1.495, 95% CI = 1.093–2.045, *p* = 0.012; occupational exposure, OR = 1.714, 95% CI = 1.253–2.346, *p* = 0.001, respectively). Occupational exposure was still a significant risk factor for future exacerbation (OR = 1.418, 95% CI = 1.027–1.956, *p* = 0.033) as well as SGRQ-c in model 2, which additionally included mCCI and SGRQ-c total score.

### 3.3. Occupational Exposure as a Contributing Factor for Annual FEV_1_ Decline

According to the LMM, adjusting for occupational exposure, sex, age, current smoking, baseline postBD FEV_1_, history of exacerbations in the previous year, BMI, emphysema index, and 6MWD, occupational exposure had a significant impact on the annual change of postBD FEV_1_ (*p* = 0.009). In addition, annual decline of FEV_1_ was also significantly affected by sex, age, baseline postBD FEV_1_, and emphysema index (Table 3).

### 3.4. Impact of Occupational Exposure on Quality of Life in COPD Patients

The LMM was analyzed in the model, including occupational exposure, female sex, age, current smoking, baseline postBD FEV_1_, history of exacerbation in the previous year, BMI, emphysema index, and 6MWD. History of exacerbation during the previous year and a higher emphysema index showed an association with worsening quality of life in COPD patients. However, the trend was observed for occupational exposure but without statistical significance (Table 4).

## 4. Discussion

In the KOLD cohort, 76 (24.4%) of 312 adult patients with COPD had experienced at least one occupational exposure. A higher risk of acute exacerbation and a significant decrease in lung function was observed during the follow-up period in COPD patients who had experienced occupational exposure. They also had lower quality of life at the time of enrollment but showed a trend toward deterioration of the quality of life at the follow-up measure. These findings are noteworthy because the results were obtained after adjusting for potential confounders that might affect them.

Besides smoking, occupational or environmental exposure as a potent risk factor for the development of COPD must be considered in never-smoking patients with COPD [12,27,28]. Lamprecht B. et al. reported that occupational exposure is a predictor of never-smoker COPD. Occupational exposure lasting more than 10 years, such as to organic dust, gases, or vapors, would increase the risk for the development of COPD [27]. According to data from the National Health and Nutrition Examination Survey III conducted on the US population, exposure in the workplace was a factor in the development of COPD in never-smokers [29]. However, most reported studies are epidemiological studies on exposure in the workplace [27,29,30], and the clinical course after occupational exposure has not been clarified.

Acute exacerbation is a detrimental event in COPD patients; various risk factors are associated with acute exacerbations of COPD [31,32]. As reported in several studies, the effect of occupational exposure, as well as a previous exacerbation within one year, on COPD exacerbation has been demonstrated [33,34]. However, in this study, while the history of exacerbation during the previous year lost its significance in the adjusted model, a history of occupational exposure was a significant risk factor in the future exacerbation of COPD. In other words, occupational exposure augmented airway inflammation and had an independent impact on the exacerbation rate.

COPD is a complex and heterogeneous disease. Exposure to noxious particles/gases and predisposing genetic factors plays a potential role in the pathogenesis of COPD [1,35]. Cigarette smoke, the most representative noxious particle activates inflammatory chemokines and cytokines in the small airways, directly causing damage to airway epithelium. Finally, it contributes to permanent structural change in the airway [36,37]. This process could also be applied to occupational noxious particles/gases. Occupational exposure in the workplace would trigger the release of chemokines or cytokines and cause chronic airway inflammation. Unresolved airway inflammation subsequently causes remodeling of the airway and parenchymal destruction. Thus, airway inflammation induced by occupational exposure could be a risk factor for decline of lung function [6,38]. In our study, we checked various materials of occupational exposure, and the causative materials can be possible of inducing the airway inflammation. Additionally, the emphysema index and occupational exposure showed an association with decline of lung function.

Unlike acute exacerbation or decline of lung function, occupational exposure showed a deteriorating trend; however, it had no significant effect on the quality of life in the follow-up. The lower quality of life observed in patients with occupational exposure at enrollment could influence this result. However, these results are consistent with findings of several previously reported studies analyzing the quality of life in COPD patients. According to the previous studies, poor quality of life showed an association with comorbidities, uncontrolled respiratory symptoms such as sputum, or pulmonary hyperinflation (represented by residual volume/total lung capacity), while no significant association was observed with regard to smoking history or airflow limitation [39,40,41].

In our study, patients with occupational exposure showed a lower level of education. The causative factors of occupational exposure in this study revealed welding solvent, wheat flour powder, grain dust, hair performance drug, etc. These came from a job related to the rural region, or a job that requires education not higher than college but professional skills.

This study has several limitations. First, subsequent occupational exposure was not examined after the investigation of occupational exposure at baseline enrollment. Second, because the occupational exposure was checked by questionnaire and the duration of the occupational exposure could not be determined, there can be a possibility of a recall bias and we could not measure the quantification of exposure or the accumulating impact of occupational exposure. Third, specific chemical agents were not identified as occupational exposure in 58 cases. These limitations were inevitable in terms of self-reported study design. Fourth, the effect of treatment on the outcome of COPD was not considered. However, because the management of most patients was based on the GOLD document in the university hospitals, there could be no differences in treatment between the two groups. Finally, we analyzed some patients whose occupational history was investigated among the patients in COPD cohort. Our findings cannot be considered to reflect the results of the entire cohort, and our study suggests that occupational exposure may influence not only the development of COPD but also the clinical course of COPD. Despite these limitations, while most published studies on COPD in patients with occupational exposure are cross-sectional studies related to the development of COPD following occupational exposure, our study investigating occupational exposure in a prospective COPD cohort is rare.

## 5. Conclusions

The findings of this study demonstrated that, even after adjusting for confounding factors, occupational exposure would be a prognostic factor for acute exacerbation and decline of lung function in patients with COPD. Therefore, clinical physicians should monitor the prognosis of COPD patients who have experienced occupational exposure.

## Figures and Tables

**Figure 1 jpm-12-01592-f001:**
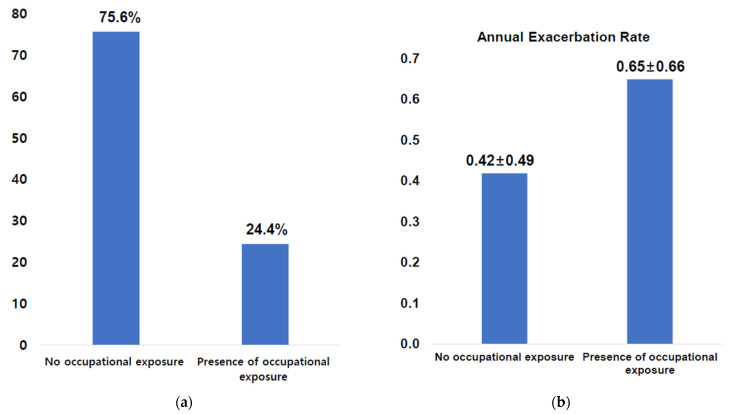
(**a**) Prevalence according to the history of occupational exposure; (**b**) annual exacerbation rate according to the history of occupational exposure.

**Table 1 jpm-12-01592-t001:** Baseline characteristics of COPD patients with and without history of occupational exposure (total *n* = 312).

Variable	Patients with Occupational Exposure (*n* = 76)	Patients without Occupational Exposure (*n* = 236)	*p*-Value *
Age, year	65.6 ± 7.6	67.0 ± 7.5	0.184
Female sex	2 (2.6)	8 (3.4)	0.740
BMI, kg/m^2^	22.6 ± 3.3	23.2 ± 3.28	0.183
mMRC dyspnea grade	1.84 ± 1.0	1.61 ± 1.0	0.105
Current smoker	24 (32.0)	76 (33.3)	0.539
Education level, college or more	8 (10.5)	54 (22.9)	0.012
Quality of life			
CAT score, total	17.4 ± 8.6	14.6 ± 8.9	0.316
SGRQ-c total	39.7 ± 18.8	33.1 ± 17.6	0.005
6MWD, m	419.9 ± 85.4	436.6 ± 86.8	0.132
BODE index	2.43 ± 2.2	2.15 ± 1.9	0.275
mCCI	0.36 ± 0.7	0.29 ± 0.5	0.398
Post-BD spirometric values			
FVC, L	3.45 ± 0.78	3.47 ± 0.79	0.844
FVC, % of predicted value	89.8 ± 18.4	91.8 ± 18.1	0.403
FEV_1_, L	1.61 ± 0.62	1.62 ± 0.53	0.901
FEV_1_, % of predicted value	59.0 ± 19.7	60.9 ± 18.4	0.440
FEV_1_/FVC ratio (%)	46.4 ± 12.4	46.8 ± 10.6	0.832
DLCO, % predicted	74.5 ± 22.8	79.6 ± 24.8	0.117
Emphysema index	25.1 ± 16.3	22.7 ± 15.4	0.335
Acute exacerbation			
Exacerbations in the previous year ≥ 1	23 (30.3)	41 (17.5)	0.017
Follow-up period after register, month	54.2 ± 22.5	56.5 ± 24.5	0.482
Presence of exacerbation during the follow-up period	48 (69.6)	148 (67.6)	0.758

Data are presented as mean ± SD or *n* (%); * *p* values for categorical variables were calculated with the χ^2^ test or Fisher’s exact test; *p* values for continuous variables were estimated with the student’s *t*-test or Mann–Whitney u-test. Definition of abbreviations: COPD, chronic obstructive pulmonary disease; CAT, chronic obstructive pulmonary disease (COPD) assessment test; SGRQ-c, COPD-specific version of the St. George respiratory questionnaire; BODE index, Body mass index, Airway Obstruction, Dyspnea, Exercise tolerance; BMI, body mass index; FEV_1_, forced expiratory volume in 1 s; mMRC, modified medical research council; 6MWD, six-minute walk distance; mCCI, modified Charlson comorbidity index; post-BD, post-bronchodilator; FVC, forced vital capacity; DLCO, diffusing capacity of the lungs for carbon monoxide.

**Table 2 jpm-12-01592-t002:** Risk factors of acute exacerbation in the two generalized estimating equation models.

	Model 1 *			Model 2 ^†^		
Factor	OR ^‡^	95% CI	*p* Value	OR ^‡^	95% CI	*p* Value
Presence of occupational exposure	1.714	1.253–2.346	0.001	1.418	1.027–1.956	0.033
Sex (female for male)	1.531	0.867–2.705	0.142	1.601	0.830–3.088	0.160
Age, an increase by 1	0.998	0.977–1.019	0.888	0.995	0.974–1.017	0.696
Smoking status (never or ex-smoking for current smoking)	0.921	0.677–1.252	0.601	0.860	0.638–1.158	0.322
BODE index, an increase by 1	1.147	1.065–1.235	<0.001	1.031	0.966–1.775	0.476
Acute exacerbation in the previous year	1.495	1.093–2.045	0.012	1.309	0.966–1.775	0.082
mCCI, an increase by 1	-	-	-	1.145	0.945–1.387	0.116
SGRQ-c total score, an increase by 1	-	-	-	1.019	1.010–1.028	<0.001

* Model 1: history of occupational exposure, sex, age, BODE index, smoking status, history of exacerbation in the previous year. **^†^** Model 2: history of occupational exposure, sex, age, BODE index, smoking status, history of exacerbation in the previous year, mCCI, SGRQ-c total score. ^‡^ OR was adjusted by the variables included in the corresponding model. Definition of abbreviations: CI = confidence interval, OR, odd ratio; BODE, Body mass index, Airway Obstruction, Dyspnea, Exercise tolerance; mCCI, modified Charlson comorbidity index; SGRQ-c, COPD-specific version of the St. George respiratory questionnaire.

**Table 3 jpm-12-01592-t003:** Factors that had impact on annual change of post-bronchodilator FEV_1_.

Factor (Baseline)	Estimated β (SE)	*p* Value *
Occupational Exposure	−0.519 (0.199)	0.009
Sex (female)	−0.499 (0.094)	<0.001
Age	−0.021 (0.002)	<0.001
Current smoking	0.006 (0.038)	0.807
Baseline postBD-FEV1	0.020 (0.001)	<0.001
History of exacerbation in the previous year	−0.015 (0.044)	0.736
BMI	0.009 (0.006)	0.119
Emphysema index	−0.003 (0.001)	0.012
6MWD	0.001 (0.001)	0.092

***** Analysis by linear mixed model adjusting all the factors in this table. Definition of abbreviations: FEV1, forced expiratory volume in 1 s; postBD-FEV1, post-bronchodilator forced expiratory volume in 1 s; BMI, body mass index; 6MWD, six-minute walk distance SE, standard error.

**Table 4 jpm-12-01592-t004:** Factors that had impact on quality of life (COPD-specific version of the St. George respiratory questionnaire) in COPD.

Factor (Baseline)	Estimated β (SE)	*p* Value *
Occupational Exposure	1.986 (2.752)	0.471
Sex (female)	9.047 (5.138)	0.080
Age	0.147 (0.139)	0.916
Current smoking	3.561 (2.088)	0.090
Baseline postBD-FEV1	−0.169 (0.056)	0.003
History of exacerbation in the previous year	5.586 (2.414)	0.022
BMI	0.290 (0.342)	0.397
Emphysema index	0.325 (0.075)	<0.001
6MWD	−0.058 (0.012)	<0.001

***** Analysis by linear mixed model adjusting all the factors in this table. Definition of abbreviations: COPD, chronic obstructive pulmonary disease; postBD-FEV1, post-bronchodilator forced expiratory volume in 1 s; BMI, body mass index; 6MWD, six-minute walk distance SE, standard error.

## Data Availability

The datasets used and/or analyzed during the current study are available from the corresponding author on reasonable request.

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
