# Peer review of "Impact of Previous Occupational Exposure on Outcomes of Chronic Obstructive Pulmonary Disease"

_jpm, 2022, doi:10.3390/jpm12101592_

Round 1

Reviewer 1 Report

My comment

Q1. In your manuscript you did not show how your exposure was measured. Please add in your manuscript how KOLD cohort assessed exposure.

Except the question, what tools are used in the KOLD cohort for evaluation, identification, and classification of the different exposures.

Q2. Why patients who had experienced occupational exposure had significantly lower level of education? Would you add the plausible hypothesis in your discussion section?

Q3. In your manuscript, you talk about several exposures, we don't know which exposure is the real responsible of COPD outcomes. As long as exposure is poorly evaluated, it is difficult to establish the relationship between exposure and COPD outcomes. What do you think about?

Q4. In your manuscript, there are no results relating to the duration of exposure and the concentration of chemicals. Also, occupational exposure with identified chemical represented only 5.76% of your sample size. I think it's difficult by a simple question of knowing if subject have exposed in the past to one of chemicals to get that conclusion in your manuscript. I need your comment.

Q5. I think that your sample is not representative because patients with identified occupational exposure chemicals represent: (18/312=5.76%) of the sample size; patients with not identified occupational exposure chemicals or other chemical exposure represent: (58/312=18.6), of the sample size; patients without occupational exposure represent :(236/312=75.6) of the sample size. Therefore, the comparison is not easy, and we can have the selection bias in the case of a cross-sectional study. What do you think about?

Q6. How did you evaluate the exacerbation rate?  Exacerbation of COPD outcomes included increased dyspnea, health status, rate of hospitalization, readmission, disease progression and symptoms. How did you classify the presence and absence of exacerbation?

Q7. How do you explain that there is no difference between the patients with occupational exposure and the patients without occupational exposure in the terms of parameters which define and/or classify the COPD: Post BD spirometry values, Emphysema index, degree of dyspnea, but the occupational exposure had a negative impact on COPD outcomes? That means occupational exposure is not a predictor for COPD but a predictor for COPD outcome in your study.

Q8. Ageing is a risk factor of COPD. In your study the average age of patients with occupational exposure is 65 years. The effect of age associated with COPD disease has a real influence on the quality of life. Do you think that occupational exposure is the only factor in your study that negatively influences the quality of life according to SGRQc?

Q9. Increased dyspnea is the key symptom of COPD in term of the frequency of exacerbation. Regarding the results of the dyspnea scale in your study, you found no difference between patients with occupational exposure and patients without occupational exposure, but in your conclusion occupation exposure is related to worse outcome in the term of exacerbation rate. What do you think about?

                                Thank so much

Author Response

## Response to Reviewer 1

Q1. In your manuscript you did not show how your exposure was measured. Please add in your manuscript how KOLD cohort assessed exposure.

Except the question, what tools are used in the KOLD cohort for evaluation, identification, and classification of the different exposures.

Response: Thank you for your valuable comment. As you commented, we also have regrets about not being measured about the exposure quantification. In our study, the occupational exposure was checked by the questionnaire, so, we could not measure the quantification of exposure or the accumulating effect of exposure. We corrected the contents in Discussion section.

Second, because the occupational exposure was checked by questionnaire and the duration of the occupational exposure could not be determined, there can be a possibility of a recall bias and we could not measure the quantification of exposure or the accumulating impact of occupational exposure.”

Q2. Why patients who had experienced occupational exposure had significantly lower level of education? Would you add the plausible hypothesis in your discussion section?

Response: Thank you for your comment. The causative factors of occupational COPD in our study revealed welding solvent, wheat flour powder, grain dust, hair performance drug, and so on. These came from a job related to the rural region, or a job that requires not higher education than college, but professional skills. Therefore, in our study, COPD patients with occupational exposure showed the lower level of education. As you recommended, we added this hypothesis in our discussion section.

“In our study, patients with occupational exposure showed a lower level of education. The causative factors of occupational exposure in this study revealed welding solvent, wheat flour power, grain dust, hair performance drug, and so on. These came from a job related to the rural region, or a job that requires not higher education than college, but professional skills.”

Q3. In your manuscript, you talk about several exposures, we don't know which exposure is the real responsible of COPD outcomes. As long as exposure is poorly evaluated, it is difficult to establish the relationship between exposure and COPD outcomes. What do you think about it?

Response: Thank you for your valuable comments. We totally agreed with your opinion. In order for us to clearly prove the causality of exposure and COPD outcome, we should have quantified the exposure amount of measured the exposure continuously. Since the exposure was checked with a self-reported questionnaire about exposure at the time of enrollment, that remains a limitation of our study. We tried to focus on the clinical course of COPD according to occupational exposure, but due to a few limitations, additional research will be needed in the future.

Q4. In your manuscript, there are no results relating to the duration of exposure and the concentration of chemicals. Also, occupational exposure with identified chemical represented only 5.76% of your sample size. I think it's difficult by a simple question of knowing if subject have exposed in the past to one of chemicals to get that conclusion in your manuscript. I need your comment.

Response: We appreciate your helpful suggestion. As you commented, unfortunately, we did not check the duration of exposure and the concentration of chemicals. There is a possibility of a recall bias by only asking questions about occupational exposure at the time of enrollment, so we described the limitation about checking the occupational exposure in our study.  

Second, because the occupational exposure was checked by questionnaire and the duration of the occupational exposure could not be determined, there can be a possibility of a recall bias and we could not measure the quantification of exposure or the accumulating impact of occupational exposure.”

Q5. I think that your sample is not representative because patients with identified occupational exposure chemicals represent: (18/312=5.76%) of the sample size; patients with not identified occupational exposure chemicals or other chemical exposure represent: (58/312=18.6), of the sample size; patients without occupational exposure represent :(236/312=75.6) of the sample size. Therefore, the comparison is not easy, and we can have the selection bias in the case of a cross-sectional study. What do you think about?

 Response: As you commented, we can have the selection bias in our study. However, whatever the chemical materials that cause occupational exposure, we think that the materials of occupational exposure activated the inflammatory cytokines and chemokines, and the cytokines or chemokines can affect the clinical course of COPD by the causative agents of occupational exposure. We added in Discussion section.

“In our study, we checked various materials of occupational exposure, and the causative materials can be possible of inducing the airway inflammation. Additionally, the emphysema index and occupational exposure showed an association with decline of lung function.”

Q6. How did you evaluate the exacerbation rate?  Exacerbation of COPD outcomes included increased dyspnea, health status, rate of hospitalization, readmission, disease progression and symptoms. How did you classify the presence and absence of exacerbation?

Response: Thank you for your careful comments. We defined the acute exacerbation according to GOLD guideline. An acute exacerbation was defined as worsening respiratory symptoms, requiring systemic steroids and antibiotics, a visit to the emergency room, or admission to a hospital. Exacerbation was documented at each hospital and traced using medical records. If the patients visited other clinics or emergency room, the name of clinics or emergency room, the reason for the visit, date of visit, and frequency were collected through the pre-structured interview sheet.

We described the content for evaluating the exacerbation in Method section.

“According to the guideline of the Global Initiative for Chronic Obstructive Lung Disease (GOLD), an acute exacerbation was defined as worsening respiratory symptoms, requiring systemic steroids and antibiotics, a visit to the emergency room, or admission to a hospital. Interviews with patients were conducted every three months during the follow-up period using the pre-structured interview sheet, which includes the question of whether they had visited the outpatient clinics or emergency rooms due to an increased amount of sputum, purulent sputum, or deterioration of dyspnea within three months. Exacerbation was documented at each hospital and traced using medical records. If the patients visited other clinics or emergency rooms, the name of clinics or emergency rooms, the reason for the visit, date of visit, and frequency were collected through the pre-structured interview sheet.”

Q7. How do you explain that there is no difference between the patients with occupational exposure and the patients without occupational exposure in the terms of parameters which define and/or classify COPD: Post BD spirometry values, Emphysema index, degree of dyspnea, but the occupational exposure had a negative impact on COPD outcomes? That means occupational exposure is not a predictor for COPD but a predictor for COPD outcome in your study.

Response: Our study was conducted by analyzing some patients whose occupational history was investigated among the patients in COPD cohort. Our findings cannot be considered to reflect the results of the entire cohort, and our study suggests that occupational exposure may influence not only the development of COPD, but also the clinical course of COPD. We described in Discussion section.

“Finally, we analyzed some patients whose occupational history was investigated among the patients in COPD cohort. Our findings cannot be considered to reflect the results of the entire cohort, and our study suggests that occupational exposure may influence not only the development of COPD, but also the clinical course of COPD.”

Q8. Ageing is a risk factor of COPD. In your study the average age of patients with occupational exposure is 65 years. The effect of age associated with COPD disease has a real influence on the quality of life. Do you think that occupational exposure is the only factor in your study that negatively influences the quality of life according to SGRQc?

Response: Thank you for your careful comments. Risk factors such as smoking and occupational materials, which have been found through several studies related to the development of COPD, are caused by exposure to various dangerous substances that cause inflammation in the airways for a considerable duration. In our study, the average age is 65 years old, which is considered a relatively young age to be defined as an elderly person in a currently growing aging society. Additionally, since the exact duration of occupational exposure for each patient has not been ascertained, we need to be cautious in interpreting.

Q9. Increased dyspnea is the key symptom of COPD in term of the frequency of exacerbation. Regarding the results of the dyspnea scale in your study, you found no difference between patients with occupational exposure and patients without occupational exposure, but in your conclusion occupation exposure is related to worse outcomes in term of exacerbation rate. What do you think about it?

Response: Thank you for your careful comment. As you commented, dyspnea would be one marker of predicting exacerbation of COPD. Our study is a prospective cohort study, and the history of exacerbation was defined through medical records using systemic steroids and/or antibiotics after visiting the hospital with worsening respiratory symptoms such as cough or sputum as well as dyspnea, therefore, there could be possible that it did not completely coincide with the dyspnea scale.

Reviewer 2 Report

The work by Kim et al., studying the 'Impact of Previous Occupational Exposure on Outcomes of Chronic Obstructive Pulmonary Disease' is overall an interesting study. Some comments to improve the study:

1. The authors should label the axes (Y axis) on the graphs is 1a and 1b.

2. To get a better understanding of the variability associated with the study the data should be plotted in a way that each patient is represented by a single dot.

Author Response

## Response to Reviewer 2

Comment: The work by Kim et al., studying the 'Impact of Previous Occupational Exposure on Outcomes of Chronic Obstructive Pulmonary Disease' is overall an interesting study. Some comments to improve the study:

Response: Thank you for your positive feedback and comments.

Comment 1: The authors should label the axes (Y axis) on the graphs is 1a and 1b.

Response: Thank you for your detail comment. As you commented, we labeled Y axis of both fig 1a and fig 1b.

Comment 2: To get a better understanding of the variability associated with the study the data should be plotted in a way that each patient is represented by a single dot.

Response: We are sorry, but we could not understand what you have commented, so would you please mind explaining it one more time?
